# Investigation Driven by Network Pharmacology on Potential Components and Mechanism of DGS, a Natural Vasoprotective Combination, for the Phytotherapy of Coronary Artery Disease

**DOI:** 10.3390/molecules27134075

**Published:** 2022-06-24

**Authors:** You-Gang Zhang, Xia-Xia Liu, Jian-Cheng Zong, Yang-Teng-Jiao Zhang, Rong Dong, Na Wang, Zhi-Hui Ma, Li Li, Shang-Long Wang, Yan-Ling Mu, Song-Song Wang, Zi-Min Liu, Li-Wen Han

**Affiliations:** 1School of Pharmacy and Pharmaceutical Science, Shandong First Medical University and Shandong Academy of Medical Sciences, Jinan 250000, China; zmc19941114@gmail.com (Y.-G.Z.); liuxiaxia0415@126.com (X.-X.L.); zytj1136091171@163.com (Y.-T.-J.Z.); doloda56@163.com (R.D.); 17862954657@163.com (N.W.); 17854111663@163.com (Z.-H.M.); myling501@hotmail.com (Y.-L.M.); wangsongsong@sdfmu.edu.cn (S.-S.W.); 2School of Pharmaceutical Science, Shanxi Medical University, Taiyuan 030000, China; 3Chenland Research Institute, Irvine, CA 92697, USA; jczong@chenland.com (J.-C.Z.); lli@chenland.com (L.L.); mwang@chenland.com (S.-L.W.); 4School of Pharmacy, Shandong University of Traditional Chinese Medicine, Jinan 250000, China; 5Chenland Nutritionals Inc., Irvine, CA 92697, USA

**Keywords:** coronary artery disease, phytotherapy, network pharmacology, angiogenesis, zebrafish, dynamics molecular docking

## Abstract

Phytotherapy offers obvious advantages in the intervention of Coronary Artery Disease (CAD), but it is difficult to clarify the working mechanisms of the medicinal materials it uses. DGS is a natural vasoprotective combination that was screened out in our previous research, yet its potential components and mechanisms are unknown. Therefore, in this study, HPLC-MS and network pharmacology were employed to identify the active components and key signaling pathways of DGS. Transgenic zebrafish and HUVECs cell assays were used to evaluate the effectiveness of DGS. A total of 37 potentially active compounds were identified that interacted with 112 potential targets of CAD. Furthermore, PI3K-Akt, MAPK, relaxin, VEGF, and other signal pathways were determined to be the most promising DGS-mediated pathways. NO kit, ELISA, and Western blot results showed that DGS significantly promoted NO and VEGFA secretion via the upregulation of VEGFR2 expression and the phosphorylation of Akt, Erk1/2, and eNOS to cause angiogenesis and vasodilation. The result of dynamics molecular docking indicated that Salvianolic acid C may be a key active component of DGS in the treatment of CAD. In conclusion, this study has shed light on the network molecular mechanism of DGS for the intervention of CAD using a network pharmacology-driven strategy for the first time to aid in the intervention of CAD.

## 1. Introduction

Coronary Artery Disease (CAD) is a cardiovascular disease with significant morbidity and mortality, especially among the elderly [1], and imposes a heavy burden on health care systems worldwide [2,3,4]. CAD is caused by excessive lipid accumulation in the vessel wall because of long-term exposure to lifestyle risk factors such as high sugar and fat. This accumulation results in reduced endothelial function, which in turn leads to stenosis and blockage [5]. A study has shown that vascular regeneration in the infarcted areas of the heart and the construction of new vascular transport channels at the onset of CAD are essential for alleviating CAD symptoms [6]. In this regard, phytotherapy has been used in China for thousands of years to regulate human health, especially in the case of cardiovascular and cerebrovascular diseases [7].

As an important part of phytotherapy, Traditional Chinese Medicine (TCM) provides us with a vast resource to develop supplements for health care and disease treatment [8]. In previous research in our laboratory, DGS was screened as a novel natural vasoprotective combination. DGS was composed of three Chinese medicinal materials, namely, Salviae Miltiorrhizae Radix et Rhizoma (Danshen), Puerariae Lobatae Radix (Gegen), and Crataegi Folium (Shanzhaye). Modern pharmacological studies have shown that Danshen can increase blood flow [9], improve circulation [10], and preserve endothelial function [11]. Danshen is the most representative herbal medicine in the Chinese medicine formula repertoire for the treatment of ischemic diseases [12]. Gegen is usually used as a “medicinal pair” with Danshen. Shanzhaye is also commonly utilized for promoting blood circulation and resolving blood stasis. All three herbs are used as dietary supplements for the prevention and treatment of diseases [13,14,15]. Further studies have reported salvianolic acid B in Danshen exerts significant anti-myocardial ischemic effects and can alleviate oxidative stress damage caused by myocardial ischemia [16]. Hyperoside, one of the important flavonoids in Gegen, is known for its vasoprotective effects [17]. However, the potential components and mechanisms of this combination have not been thoroughly reported.

TCM exerts a multi-component, multi-target, and multi-pathway synergistic effect [18]. However, this complexity is a bottleneck for revealing its modern scientific significance. Network pharmacology is a new approach that analyzes the complexities of drugs and diseases and visualizes the drug treatment mechanisms through the construction of biological association networks, which are in line with TCM philosophy [19]. Therefore, this study aimed to devise a new network pharmacology-driven strategy to investigate the active components and network molecular mechanism of DGS. Moreover, transgenic zebrafish were used as the in vivo experimental animal model to evaluate the bioactive effects of DGS samples. Finally, this research aspired to reveal the potential components and pharmacodynamic mechanisms of DGS in the intervention of CAD.

## 2. Results

### 2.1. Identification of the Chemical Components of DGS

HPLC-qTOF-MS/MS was used to analyze the composition of DGS. Based on chromatographic retention times and fragment ion information of the molecules, 37 compounds were identified from the DGS samples (Figure 1). The compound names, retention times (min), and molecular formulae are listed in Table 1. The *m/z* 359.0761 in the negative ion mode corresponds to the [M-H]^−^ ion peak of Rosmarinic acid. The primary and secondary mass spectra of Rosmarinic acid are consistent with the cleavage of primary and secondary given in the MoNA database. Furthermore, a molecular ion peak was seen at *m/z* 609.1415, which corresponded to the [M-H]^−^ ion peak of rutin [20]. The secondary mass spectrum of rutin showed a mass spectral fragment of *m/z* 301, a [M-H-C_12_H_22_O_10_]^−^ mass spectral fragment formed by the loss of two sugar rings of rutin. Salvianolic acid B [21], tanshinone IIA, cryptotanshinone, and hyperoside were respectively identified by the [M-H]^−^ ion peak at *m/z* 717.1220, *m/z* 293.1192, *m/z* 295.1362, and [M-H]^−^ ion peak at *m/z* 463.0851 in the negative ion mode. Tanshinone IIA, cryptotanshinone, and hyperoside were identified through the NoMB database. Based on mass spectrometry information of primary and second ion fractions, nearly half of the 34 compounds identified from the DGS samples were found to be flavonoids and their glycoside derivatives and the rest were phenolic acids and tanshinones.

### 2.2. Key Component and Target Screening and Prediction Using Network Pharmacology Analysis

A total of 35 compounds were successfully predicted to be the targets of action in the SwissTargetPrediction database. The compounds Piscidic acid and Schaftoside could not be predicted in the SwissTargetPrediction database based on two-dimensional (2D) and three-dimensional (3D) similarities. Hence, they were excluded. A total of 442 biological targets were obtained by combining these 35 compounds and removing duplicates. Moreover, 859 CAD-related targets were obtained from the Genecards database using the correlation score of >20 as the screening criterion. A Venn diagram analysis of the compound’s target library against the disease’s target library yielded 112 incorporative targets (Figure 2A). The top five enriched target categories were protease, enzyme, kinase, nuclear receptor, and secreted protein. As shown in Figure 2B, the top five targets in the protein–protein interaction (PPI) network in terms of degree-value were ALB, TNF, VEGFA, EGFR, and CASP3. Cytoscape 3.9.1 was employed to calculate the network topology data, and the MCODE plug-in was used to calculate the most closely linked clusters in the PPI network. A total of 4 clusters were obtained, with cluster scores of 23.667, 8.147, 3.5, and 3 (Figure 2B’). Cluster 1 with the highest clustering score was used for subsequent analysis.

In the biological process enrichment analysis, Cluster1 was imported into the David database for GO enrichment analysis. Annotations of genes were obtained by using GO analysis. The screening was performed on a *p* < 0.05 basis, and the top 30 biological processes were listed according to the magnitude of the *p*-value (Figure 2C). It chiefly involved cytokine-mediated signaling pathway, positive regulation of MAPK cascade, positive regulation of cell migration, positive regulation of kinase activity, protein autophosphorylation, positive regulation of MAP kinase activity, MAPK cascade, positive regulation of protein phosphorylation, positive regulation of nitric oxide biosynthetic process, positive regulation of ERK1 and ERK2 cascade, positive regulation of cell proliferation, positive regulation of phosphatidylinositol 3-kinase signaling, and negative regulation of endothelial cell apoptotic. These biological processes were mostly involved in the regulation of cytokines, cascade activation of the MAPK pathway, positive regulation of protein phosphorylation, positive regulation of NO synthesis, positive regulation of cell proliferation, anti-endothelial cell apoptosis, etc. The top 30 KEGG signaling pathways at *p* < 0.05 and sorted according to count value size. As shown in Figure 2D, KEGG pathway enrichment analysis revealed that most of the DGS targets were concerned with the regulation of cell proliferation, anti-apoptosis, cell migration, and angiogenesis.

### 2.3. Pharmacological Mechanism Analysis in HUVECs

#### 2.3.1. Effect of DGS on the Proliferation and Migration of HUVECs

The CCK-8 assay was performed to evaluate the cell viability after the treatment with DGS. As illustrated in Figure 3A, DGS significantly increased the viability of HUVECs. Six concentration groups of DGS were used to treat the HUVECs for 24 h. The mean cell survival rate of the 1 μg/mL DGS group was 116%, which did not show a proliferative effect compared with the control (Ctrl) group (100%). After treatment with DGS, the cell survival rates were 129.0% (*p* < 0.01), 127.4% (*p* < 0.01), 126.3% (*p* < 0.01), 128.0% (*p* < 0.01), and 124.3% (*p* < 0.01) in the five concentration groups of 2, 4, 8, 16, and 32, respectively, in comparison with the control group. These results indicate that DGS significantly promotes the proliferation of HUVECs.

Endothelial cell migration is essential for angiogenesis. Therefore, the effect of DGS on the migration of HUVECs was evaluated using wound-healing assays. As shown in Figure 3A,B, VEGF significantly increased the migration of endothelial cells to the wound healing zone after injury (*p* < 0.0001). Treatment of HUVECs with DGS revealed that DGS significantly promoted the migration of endothelial cells toward the middle of the scratch and promoted scratch healing.

The formation of blood vessels rests on the proliferation and migration of endothelial cells. Therefore, in addition to the above wound healing assay that examined the lateral migration of the endothelial cells, the transwell assay that investigated the longitudinal migration of the cells after the treatment with DGS was performed. As presented in Figure 3D,F, the migration of HUVECs was significantly increased compared with the control after treatment with VEGF-positive drugs. Moreover, DGS significantly increased the ability of HUVECs to migrate toward the backside of the membrane. Compared with the Control group (0.6568 HPF), the mean values of the number of cells migrating to the back of the membrane with 4, 8, and 16 μg/mL of DGS were 1.198 (*p* > 0.05), 1.763 (*p* < 0.01), and 1.946 (*p* < 0.001) HPF, respectively.

Angiogenesis was evaluated in terms of endothelial cell migration. Matrix gel was used to evaluate the ability of DGS to promote tube formation in HUVECs. Furthermore, the ability to promote tube formation was assessed based on the capillary length and the number of branching points. HUVECs treated with VEGF resulted in a mean number of branch points and a mean capillary length of 52.80 (*p* < 0.01) and 8144 μm (*p* < 0.05), respectively, compared with the control group; the mean value of the number of branch points for the three concentrations after the treatment of HUVECs with DGS control were 49.91, 65.75, and 50.64, respectively, and the lumen lengths were 8417, 10,568, and 8098, respectively (Figure 3E,G,H).

#### 2.3.2. NO and VEGF Release and Protein Expression of DGS

Compared with the control group, the NO level was increased in the high concentration group of DGS (16 μg/mL) (*p* < 0.05), but there was no difference in NO secretion between 4 μg/mL DGS and 8 μg/mL DGS groups (*p* > 0.05). In ELISA, compared with control, the secretion of 8 μg/mL DGS and 16 μg/mL VEGFA treatment groups compared to the control group (*p* < 0.05, *p* < 0.05) (Figure 4A,B).

The molecular mechanisms associated with the promotion of angiogenesis by DGS were elucidated. Further experiments to explore the pathways by which DGS enhanced the proliferation and migration of HUVECs. As shown in Figure 4C,D, DGS significantly upregulated VEGFR2, thereby activating the phosphorylation of proteins related to downstream signaling pathways. As illustrated in Figure 4C,E, the level of Akt phosphorylation was significantly elevated after DGS treatment, and the value of p-Akt/Akt was statistically different compared with that of the control group (*p* < 0.01). Furthermore, Erk phosphorylation level was found to be significantly elevated and statistically different after DGS treatment (*p* < 0.05, *p* < 0.01) (Figure 4C,F). Finally, vasodilation-related proteins were examined, which revealed that the phosphorylation levels of eNOS were also significantly elevated, with statistical differences (*p* < 0.01, *p* < 0.01) (Figure 4C,G). The increase in phosphorylation of VEGFR2 and eNOS was consistent with the increased secretion of VEGFA and NO in the cell cultures described above. These results suggest that DGS activates the VEGFA/VEGFR2/Akt/Erk/eNOS signaling pathway to promote angiogenesis and vasodilation.

### 2.4. Bioactivity Evaluation of DGS Based on Zebrafish Model In Vivo

A zebrafish assay was performed according to the schematic diagram (Figure 5A). As shown in Figure 5B, zebrafish treated with DGS for 3 days demonstrated mortality and abnormalities at concentrations ≥ 100 μg/mL. The LD50 of DGS was 753.7 μg/mL and the 5% lethality was 478 μg/mL. These results suggest that efficacy assessment of DGS at concentrations ≤ 100 μg/mL can be used for the evaluation of the efficacy of angiogenesis.

Transgenic Tg (flk1a: EGFP) zebrafish embryos were used to explore the proangiogenic effects of DGS. The findings indicated that PTK787 significantly inhibited the formation of intersegment vessels (ISVs). Most vessels were incomplete and discontinuous after PTK787 treatment (Figure 5B,D). The ISVs length in the model was 323.3 ± 38.42 μm, which was significantly lower than that of the control (1017 ± 58.96 μm; *p* < 0.0001). After treatment with CoQ10 as a positive drug, the total ISV length reached 730.0 ± 59.77 μm, which was significantly higher than that of the model group (*p* < 0.0001). The lengths of ISVs after treatment with DGS at concentrations of 10, 25, and 40 μg/mL were 610.0 ± 85.08 μm (*p* < 0.01), 793.7 ± 23.71 μm (*p* < 0.0001), and 654.3 ± 70.81 μm (*p* < 0.0001), respectively. Concerning the number of vascular sprouting, the number was significantly increased in the CoQ10 and DGS-treated groups. In conclusion, DGS was able to significantly promote the growth of the inhibited vessels.

To further investigate the proangiogenic effect of DGS on normal blood vessels, the effects were assessed using the main subintestinal veins (MSIV) of the zebrafish. As depicted in Figure 5C,F, the length of MSIV in the zebrafish treated with CoQ10 for 24 h was 688.2 ± 22.13 μm, which was significantly higher compared with that of the control (628.0 ± 22.13 μm; *p* < 0.05). After treatment with DGS at concentrations of 5, 10, and 20 μg/mL, the lengths of MSIV were 708.0 ± 18.22 μm (*p* < 0.01), 771.5 ± 22.72 μm (*p* < 0.0001), and 775.5 ± 23.62 μm (*p* < 0.0001), respectively.

### 2.5. Dynamic Molecular Docking of Active Components of DGS with Key Targets

The binding ability of compounds from DGS to VEGFR2 was validated in the present study using the same molecular docking to provide evidence for the proangiogenic effect of DGS with a virtual approach. All 37 components of DGS were docked to the binding pocket of VEGFR2, and an agonist of VEGFR2 was used as the positive control. All compounds were docked into the active pocket of VEGFR2, and all exhibited binding energies were greater than −5 kcal/mol. When the binding energy of agnuside was used as a control [22], nine compounds had binding energies greater than that of agnuside, as shown in Table 2. The binding modes of agnuside and salvianolic acid C with VEGFR2 were visualized separately using Pymol, which showed that the binding of agnuside to VEGFR2 was mainly via hydrogen bonding and pi-sigma interactions (Figure 6A). On the contrary, salvianolic acid C displayed pi–pi stacked interactions in addition to hydrogen bonding and pi–sigma interactions (Figure 6B).

Based on the docking results, salvianolic acid C and agnuside with VEGFR2 were further selected for MD simulation to examine their stability in the binding pocket. Root mean square deviation (RMSD) serves as an important basis for measuring system stability. In this study, the mean RMSD of the salvianolic acid C–VEGFR2 system after balance was 2.63 ± 0.323 Å (Figure 7A). Meanwhile, the mean RMSD of the agnuside–VEGFR2 system after balance was 2.36 ± 0.44 Å (Figure 7A). The mean RMSD of the two systems is <3 Å, which is completely acceptable in the protein system. Subsequently, the flexibility changes and root mean square fluctuation (RMSF) values of amino acid residues in VEGFR2 were evaluated. Figure 7B,F, the amino acid residues that interact with the ligand for more than 30% in the RMSF diagram. The RMSF values of most of the amino acid residues involved in the interaction were small, thereby indicating that the stability of the entire system was increased after salvianolic acid C and agnuside are combined with VEGFR2. The 2D visualization analysis of the salvianolic acid C–VEGFR2 system showed that salvianolic acid C formed hydrogen bonds with VEGFR2 via LYS-920, ASN-923, ARG-842, CYS-919, GLU-917, and GLU-885; furthermore, it forms Pi–Pi stacking interaction with PHE-1047 (Figure 7C,D). The 2-D visual analysis of the agnuside–VEGFR2 system showed that agnuside formed hydrogen bonds with VEGFR2 through VAL-914, THR-916, GLU-885, VAL-899, ASP-1046, and ILE-1025; moreover, it formed π–cation stacking with LYS-868 (Figure 7G,H). All the interaction durations exceed 30% of the whole simulation time.

## 3. Discussion

It is well known that TCM has complex component systems, and their mechanisms are often difficult to explain using existing techniques. This difficulty greatly limits the globalization of traditional Chinese medicine. Effective study of the holistic effects of complex component systems has become a huge challenge. However, “Network pharmacology” has offered a new way to solve this problem [23,24]. Using system biology and computer technology, network pharmacology builds a “disease–gene–target–drug” interaction network, and systematically displays the impacts of drugs on the disease network [25]. Therefore, compared with the conventional single-target mode of action, network pharmacology is more suitable for studying the complex effects of TCM and has been rightly called the “next-generation drug development model”. However, most of the network pharmacology studies in TCM are based on the results of network pharmacological analysis to explain the complexity of Chinese medicine, and there is a lack of mature theoretical guidance for subsequent experimental studies. Hence, this study proposes a Network pharmacology-driven strategy to investigate the molecular mechanism of TCM, with the hope of enhancing the precision of experimental research.

In this study, most of the components of DGS, including phenols, tanshinones, and flavonoids, were identified using HPLC-qTOF/MS. The phenolic compounds in Danshen have been reported to promote myocardial ischemic angiogenesis [26]. Mass spectrometry asserted the presence of salvianolic acid A, B, C, L, and other phenolic acids in DGS. Flavonoids are present in almost all plants and studies have shown that they can inhibit cardiac injury through a variety of mechanisms [27]. Mass spectrometric detection revealed the presence of rutin, hyperoside, and other flavonoids in DGS. The mechanisms by which these components acted in the treatment of CAD were investigated. Network pharmacology was used to analyze the action network of DGS, which indicated that the targets of DGS action are mainly enriched in biological processes related to cytokine regulation, protein phosphorylation, activation of the MAPK signaling pathway, and cell proliferation. Subsequently, KEGG pathway enrichment analysis revealed that the highest enrichment of targets was in the PI3K-Akt signaling pathway. A study has reported that the PI3K-Akt pathway plays a key role in the emergence, progression, and treatment of CAD, thereby activating downstream pathways that control cell survival, proliferation, migration, and other biological processes after receiving intracellular and extracellular feedback [28]. As an effector downstream of Akt, eNOS can be activated to control endothelial cell survival, proliferation, apoptosis, and intravascular environment stability after the development of coronary lesions in the heart [29,30]. Additionally, activation of downstream eNOS contributes to myocardial angiogenesis, which is similar to the effect of VEGF in promoting therapeutic angiogenesis in CAD [31]. It has been reported that Danhong injection can promote the growth of vessels in the myocardial infarct site by activating the ERK signaling pathway [32]. Therefore, the phenotypic and signaling pathway mechanisms were validated in subsequent studies.

The zebrafish is an emerging internationally recognized model animal. During zebrafish embryonic development, angiogenesis can be visualized under a microscope and is currently considered one of the best models for studies on angiogenesis [33,34]. Specifically, the transgenic Tg (flk1a:EGFP) zebrafish with green fluorescence of the vascular can be observed via video and imaged under a fluorescence microscope with great convenience to assess the effect of the drug on vessel growth [35]. All mammalian and zebrafish endothelial cells are extremely plastic and retain their plasticity even after reaching adulthood. After enduring cardiac injury, mice can form new coronary arteries from pre-existing endothelial cells [36,37]. The zebrafish vascular growth model is ideally suited as it mimics human vascular disease models for new drug screening and evaluation [38]. Our experimental results established that DGS can promote vascular growth in vivo by increasing the vessel length and the number of vascular outgrowths, without any effects on zebrafish development, at the effective dose.

The findings from network pharmacology and zebrafish experiments suggest that effective angiogenesis can enhance blood perfusion in the case of CAD and alleviate cardiac injury. Therefore, the angiogenic mechanisms of DGS in vitro were further investigated. Endothelial cells are involved in angiogenesis at the lesion site mainly via two pathways: proliferation and migration [39]. The CCk-8 assay demonstrated that DGS significantly increased the viability of HUVECs, which established that DGS could promote the proliferation of HUVECs. The ability of DGS to promote the migration of HUVECs was then assessed using the wound healing assay [40], transwell assay [41], and tube formation assay [42]. The results showed that DGS significantly stimulated the migration of HUVECs towards the scratch region and the lower transwell compartment and significantly stimulated the tube formation ability of HUVECs.

Furthermore, the levels of VEGFA and NO secretion in HUVECs were measured, and angiogenic and diastolic-associated proteins were quantified using Western blot. NO is a key cell-signaling molecule that is produced by activated eNOS catalyzing L-arginine[43]. Activation of eNOS and release of NO will induce the vasodilation of vessels, and NO is necessary to maintain endothelial cell function [44]. In this study, the secretion of NO and the phosphorylation level of eNOS in HUVECs were measured, which signified that DGS significantly upregulated p-eNOS and stimulated the secretion of NO. Moreover, NO is one of the main mechanisms that promote angiogenesis [45]. Our results suggest that DGS can stimulate the secretion of VEGFA to upregulate VEGFR2, thereby accentuating the binding of VEGFA to VEGFR2. ERK and Akt are located downstream of the VEGF/VEGFR2 pathway, and it has been shown that the phosphorylation of Erk and Akt can further mediate the proliferation and migration of HUVECs, thereby promoting angiogenesis [46]. The results allude that DGS may promote angiogenesis and vasodilation by promoting the phosphorylation of Akt and Erk. As shown in Figure 8, DGS activates the downstream Akt/Erk/eNOS pathway by promoting the binding of VEGFA to VEGFR2. This activation ultimately releases NO to expand blood vessels and mediates the proliferation, migration, and angiogenesis of HUVECs. The dynamic molecular docking results showed that the key compounds in DGS exhibit a good binding capacity for VEGFR2, specifically salvianolic acid C, which may be an important agonist of VEGFR2 and activate downstream signaling pathways.

## 4. Materials and Methods

### 4.1. Cell and Reagents

HUVECs were purchased from Jinan Dinguo Changsheng Biotechnology Co., Ltd. (Jinan, China). PTK787 was purchased from the AbMole BioScience (Houston, TX, USA). Pronase E and dimethyl sulfoxide (DMSO) were purchased from Sigma-Aldrich (Mannheim, Germany). Coenzyme Q10(CoQ10) was purchased from Yuanye (≥98% HPLC, Shanghai, China). Methyl alcohol and acetonitrile were purchased from TEDIA (Anhui, China). Cell Counting Kit-8 (CCK-8) and bovine serum albumin (BSA) were purchased from GeneView (Houston, TX, USA). Nitric oxide (NO) assay kit was purchased from Nanjing Jiancheng Bioengineering Institute (Nanjing, China). A Human VEGF ELISA kit was purchased from ABclonal (Boston, MA, USA). We used the following primary antibodies: rabbit anti-GAPDH (Cell Signaling; Danvers, MA, USA), rabbit anti-VEGFR2 (Cell Signaling), rabbit anti-Akt (Cell Signaling), rabbit anti-p-Akt (Cell Signaling), rabbit anti-Erk1/2 (Cell Signaling), rabbit anti-p- Erk1/2 (Cell Signaling), rabbit anti-eNOS (Cell Signaling), and rabbit anti-p-eNOS (Cell Signaling). We used the following secondary antibody for the experiment: HRP-conjugated goat anti-rabbit IgG (ABclonal).

### 4.2. Plant Material and Extraction

The DGS is composed of Danshen/Gegen/Hawthorn leaf = (1:1:1. Ratio). The medicinal material samples of Danshen, Gegen, and Hawthorn leaves were collected from Hongji Tang Herbal Drug Store (Qingdao, China) and identified as Salviae Miltiorrhizae Radix et Rhizoma (the root of Salvia miltiorrhiza Bge.), Puerariae Lobatae Radix (The root of Pueraria lobata (Willd.) Ohwi), and Crataegi Folium (the leaf of Crataegus pinnatifida Bge.) by Dr. Liwen Han, Shandong First Medical University and Shandong Academy of Medical Sciences. The sample preparation method was as follows: 1000 g of crushed Danshen was extracted twice with 10 L of 80% ethanol at reflux for 2 h each time, filtered, and then combined with the extract, and concentrated and dried under reduced pressure to obtain the Danshen extract. Then, 1000 g of Gegen was crushed and extracted thrice with 10 L of 70% ethanol for 2 h each time; the extracts were filtered, concentrated, and then dried under reduced pressure. The Hawthorn leaves were then crushed and extracted by refluxing twice with 70% ethanol, each time for 2 h; the obtained extract was filtered, combined, concentrated, and then dried under reduced pressure to obtain the Hawthorn leaf extract. Three extracts were blended to obtain the sample of DGS.

### 4.3. Chromatographic and Mass Spectrometry Conditions

The DGS powder sample was dissolved in methanol at the concentration of 6.6 mg/mL methanol solution and filtered through a 0.22 μm microporous membrane. SCIEX ExionLC^TM^ AD ultra-performance liquid chromatography system, and SCIEX X500R Quadrupole time-of-flight mass spectrometer (SCIEX, Framingham, MA, USA) with SCIEX OS software were used under the chromatographic condition: Shimadzu Shim-pack GIST-C18 column (2 um, 3.0 × 100 mm), column temperature 40 °C; mobile phase composed of 0.1% formic acid in water and methanol in B; the gradient elution program 0 min, 18% B; 1 min, 18% B; 10 min, 35% B; 27 min, 55% B; 35 min, 80% B; 48 min, 82% B; 49 min, 18% B. The flow rate was set to 0.4 mL/min, the injector temperature was set to 15 °C, and the injection volume was 2 μL.

Mass spectrum condition: scanning was performed in the negative ionization mode using an electrospray ionization (ESI) source. The data acquisition method was information dependent acquisition (IDA). The scan range was 50–2000 Da, the ion source temperature was 550 °C, the declustering voltage was 5500 V and the accumulation time was 0.2 s. The settings for Gas 1, Gas 2, Curtain Gas, and CAD Gas were 50, 55, 35, and 7 psi, respectively. The data were processed using the SCIEX OS software. Compounds were characterized using primary and secondary mass spectra plotted by comparison with literature or the MassBank of North America Database (MoNA) (https://mona.fiehnlab.ucdavis.edu/, accessed on 10 November 2021).

### 4.4. Collection of DGS-Related Targets and CAD-Related Targets

The compounds’ structure obtained by mass spectrometry was input in the SMILES format using the Pubchem (https://pubchem.ncbi.nlm.nih.gov/, accessed on 10 December 2021) database. The compounds were imported into the SwissTargetPrediction (http://www.swisstargetprediction.ch/, accessed on 10 December 2021) database [47] in SMILES format, and the targets of Homo Sapiens species were selected. The results of the predicted targets were screened with a probability value > 0. Finally, the predicted targets were combined and de-duplicated to build a compound target library. All targets for the disease were obtained in the GeneCards (https://www.genecards.org/, accessed on 10 December 2021) database [48] using the keyword “Coronary Artery Disease“ and the targets were filtered according to the Relevance score > 20.

### 4.5. Protein–Protein Interaction (PPI) Network

The intersecting targets were obtained by combining the DGS target library with the CAD target library using the online Venn Analysis tool (http://bioinformatics.psb.ugent.be/webtools/Venn/, accessed on 10 December 2021). All intersecting targets were imported into the String database to obtain protein interaction network data for the intersecting targets. The obtained protein data network was imported into Cytoscape 3.9.1 for PPI visualization and topological analysis. MOCODE, a data connectivity-based plug-in for finding dense regions of the protein interaction networks, was employed to uncover the most closely related protein interaction networks in a complex network [49]. The PPI network was further analyzed using the MCODE plugin in Cytoscape 3.9.1 to identify small, closely related networks in the entire interactions, and the targets in the interactions with the highest scores were selected for subsequent enrichment analyses based on the MCODE clustering scores.

### 4.6. Gene Ontology (GO) and KEGG Pathway Enrichment Analysis

The targets obtained from the MCODE plugin screening were imported into the David (https://david.ncifcrf.gov/ accessed on 20 June 2022) database [50,51] for GO enrichment analysis and the KEGG pathway enrichment analysis. The top 30 enrichment results were visualized using Tableau2019 using either *p*-value or Count value; *p* < 0.05 items were retained for the screening results.

### 4.7. Cell Culture and Treatment

HUVECs were cultured at 37 °C in DMEM (Gibco; Thermo Fisher Scientific, Inc., Waltham, MA, USA) with 10% fetal bovine serum (FBS, Gibco; Thermo Fisher Scientific, Inc., Waltham, MA, USA), 1% penicillin–streptomycin (Gibco), and 1% endothelial cell growth supplement (ECGS). The culture medium was refreshed every other day unless otherwise stated. The drugs used to treat HUVECs cells were formulated with DMSO, and the amount of DMSO in all cell cultures was required to not exceed 5‰ of the total volume.

### 4.8. CCK-8 Assay

CCK-8 was used to test the viability of cells after treatment with different concentrations of DGS. The HUVECs were seeded into a 96-well plate at a density of 1 × 10^4^ cells per well. The microplate was placed in a CO_2_ incubator at 37 °C for 24 h, after which 10 μL of the CCK-8 solution was added to each well and the plate was incubated for 2 h. The absorbance was measured using an enzyme marker (Spectramax id5) at 450 nm.

### 4.9. Wound Healing Assay

HUVECs cells were seeded in a 6-well plate and cultured overnight until a confluent monocytic layer was formed and a straight cell scratch was made on the monolayer with the tip of a 200 μL pipette tip. The cells were treated with VEGF or different concentrations of DGS. The scratches were imaged separately using an Olympus IX83 inverted microscope (10×, magnification) (Olympus, Tokyo, Japan). The scratch area was quantified using Image Pro Plus.

### 4.10. Transwell Assay

In this study, the migration and invasive abilities of cultured HUVECs were assayed using the Transwell culture system (Corning^®^). To the lower chamber of the Transwell, DMEM medium containing the drug but without FBS. HUVECs cells were placed in the upper chamber of the DMEM medium without FBS at 37 °C for 24 h. After treatment, the cells were fixed with 4% paraformaldehyde for 15 min, and stained with 0.5% crystal violet for 10 min. Cell migration was imaged and quantified using an Olympus IX83 inverted microscope.

### 4.11. Tube Formation Assay

The angiogenic potential of HUVECs was assessed by in vitro tube formation assays. Briefly, a 96-well plate was pre-coated with Matrigel (Corning^®^) substrate (50 μL/well) and then polymerized at 37 °C for 60 min. Approximately 3 × 10^4^ HUVECs were seeded in each well and the plate was incubated in a CO_2_ incubator for 6–8 h at 37 °C. Finally, the tubular structures formed by HUVECs in the 96-well plate were observed under an inverted microscope.

### 4.12. VEGF and NO Level Detection

Cultures of drug-treated HUVECs cells were collected separately, and the supernatant was removed through centrifugation at 10,000 rpm and then stored at −20 °C. The VEGFA expression was assessed using an ELISA kit (R&D), as per the manufacturer’s institutions. The amount of NO secreted was enumerated according to the manufacturer’s instructions.

### 4.13. Western Blotting

Cells in the exponential differentiation phase were diluted to a cell suspension of 1 × 10^5^ cell/mL using complete medium. Cells were seeded into a 6-well plate at 2 mL/well in an incubator at 37 °C and 5% CO_2_. When the cells spread across 70% of the bottom of the 6-well plate, the complete medium was replaced with serum-free medium with or without DGS drug for 24 h.

HUVECs cells were washed and scraped out after centrifugation at 4 °C, followed by washing in RIPA buffer (150 mM NaCl, 50 mM Tris-HCl, 1% Triton, 0.5% NP40, 1 nM PMSF). After centrifugation at 10,000 rpm for 30 min, the supernatant was collected. The protein concentrations were determined using the BCA Protein Assay Kit (Beyotime Biotechnology, Shanghai, China). Each lane was loaded with 20 µg of the protein and separated. The proteins were then loaded on each lane and separated on SDS-PAGE gels, followed by electrophoretic transfer to nitrocellulose (NC) membranes (MilliPore, Boston, MA, USA). The membranes were blocked with 5% BSA for 2 h and then incubated with primary antibodies at 4 °C. The primary antibodies were diluted in the BSA buffer, and secondary antibodies were diluted in TBST. The antibody-reactive bands were revealed by chemiluminescence. The images were scanned and band intensities were analyzed with Image J software.

### 4.14. Animals

Adult wild-type (AB) zebrafish and Tg (flk1a:EGFP) transgenic zebrafish (labeled green fluorescent protein at endothelial growth factor receptor) were reared and managed at the Zebrafish Research Center in the School of Pharmacy and Pharmaceutical Sciences, Shandong First Medical University. The fish were kept in a recirculating system (Shanghai Haisheng Biological Experimental Equipment Co., Ltd., Shanghai, China) at a temperature of 28 °C under a mixed light schedule of 12 h light/dark. The water systems were monitored for nitrite (<0.2 ppm), nitrate (<50 ppm), and ammonia nitrogen (0.01–0.1 ppm). Conductivity and pH were maintained at 500 µS cm^−1^ and 7, respectively. The fish were fed with fungus shrimp twice daily and fasted for a day before spawning. The day before spawning the fish were placed in spawning tanks in a 1:1 ratio of male to female; the next day the isolation plates were removed after light stimulation for natural spawning, and all eggs were collected and placed in the E3 culture water (5 mM NaCl, 0.17 mM KCl, 0.4 mM CaCl_2_, 0.16 mM MgSO_4_) for incubation.

### 4.15. Animal Grouping and Administration

Transgenic Tg (flk1a: EGFP) zebrafish embryos (24 hpf) were stripped of their egg membranes in Petri dishes containing Pronase E (1 mg/mL) and randomly distributed in 24-well plates with 10 strips/well, using 3 replicates set up for each concentration group. The plates were then treated with different concentrations of DGS, the specific concentrations administered are illustrated in the corresponding results. After 72 h of treatment at 28.5 °C, the development and mortality of each group of zebrafish were enumerated.

Transgenic Tg (flk1a: EGFP) zebrafish embryos (24 hpf) were stripped of their egg membranes with Pronase E (1 mg/mL) in Petri dishes, which were randomly distributed in 6-well plates and then treated with different concentrations of DGS as indicated in the corresponding results. After treatment at 28.5 °C for 24 h, the zebrafish larvae were rinsed several times with water and then transferred to a 96-well plate. The total length of the intersegment vessels (ISVs) of zebrafish larvae was observed and imaged under a fluorescent microscope. The total ISV length was quantified for each zebrafish using the Image Pro Plus and the average ISVs length was calculated for each group of zebrafish larvae.

Transgenic Tg (flk1a: EGFP) zebrafish larvae (72 hpf) were randomly distributed in 6-well plates and then treated with different concentrations of DGS, as indicated in the corresponding results. After treatment at 28.5 °C for 24 h, the zebrafish larvae were rinsed several times with water and then transferred into a 96-well plate with the appropriate amount of anesthetic. The total length of the main subintestinal vein (MSIV) of zebrafish larvae was determined under a fluorescent microscope. The total MSIV length of each zebrafish was quantified using Image Pro Plus, and the mean MSIV length was calculated for each group of zebrafish larvae.

### 4.16. Dynamics Molecular Docking

The molecular docking study was conducted with the PyRx v0.8 [52]. VEGFR2 (PDB ID:3B8Q) was obtained from the RCSB Protein Data Bank (https://www.rcsb.org/, accessed on 1 March 2022). All polar hydrogens of the protein crystals were added, and the solvent water molecules were removed and converted into the pdbqt format. All compounds were obtained from the PubChem database as the sdf format files, and saved in the pdbqt format after conversion to an energy-minimized form using Pthe yRx v0.8. Docking was performed in the PyRx v0.8 with the following docking box parameters center_x 39.0 center_y 33.4 center_z 14.5 size_x 19.4 size_y 27.7 size_z 17.4. The results were visualized with the Pymol v2.4.1.

Desmond software package (developed at D. E. Shaw Research) was used to investigate the molecular interactions. In the molecular dynamics simulations, the complexes were placed into an automatically calculated cube box in which the complexes were modeled separately using transferable interatomic potential with three points model (TIP3P). The optimization of the models was further accomplished by optimized potentials for liquid simulations 4 (OPLS4). The system was neutralized by the addition of NaCl to make the system isotonic. A Nosehoover thermostat was used to provide a temperature of 300 k. The Martyna–Tobias–Klien barostat was used to maintain a pressure of 1.01325 bars. The total time for the molecular dynamics simulation was 50 ns. Ligand–protein interactions were simulated using the Interaction Diagram tool in the Desmond package. The Desmond package was used to generate the RMSD and RMSF of the proteins, and the interactions were further analyzed.

### 4.17. Statistical Analysis

Data are presented as the mean ± SEM. All statistical analyses were performed using the GraphPad Prism 9.0 software. The Student’s two-tailed *t*-test was performed to compare the two study groups, while one-way ANOVA was applied to compare multiple groups. *p* < 0.05 was considered to indicate statistical significance.

## 5. Conclusions

We explored the potential mechanisms of DGS as a phytotherapy for CAD by employing a network pharmacology-driven strategy. Our findings suggest that DGS may exert proangiogenic and vasodilatory effects through the activation of the VEGF/VEGFR2/Akt/Erk/eNOS signaling pathway. Molecular docking and molecular dynamics suggest that salvianolic acid C may be a key component in exerting angiogenic and vasodilatory effects. Furthermore, our study results can serve as a reference for the mechanism of DGS as a natural product for phytotherapy.

## Figures and Tables

**Figure 1 molecules-27-04075-f001:**
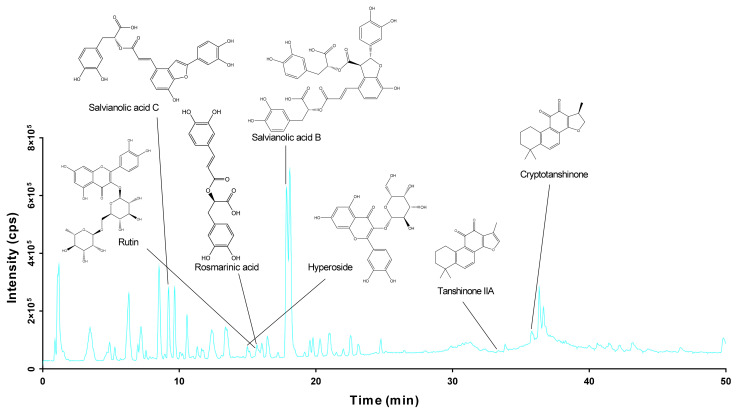
The HPLC-QTOF/MS total ion chromatogram of DGS in the negative ion modes.

**Figure 2 molecules-27-04075-f002:**
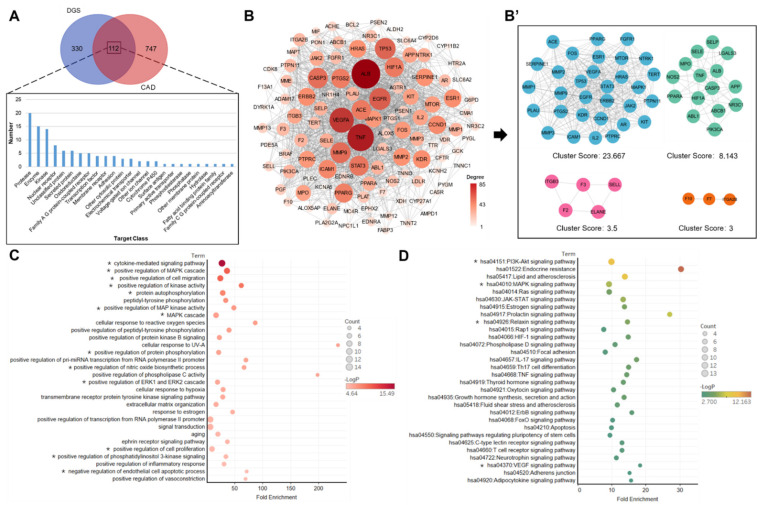
Network pharmacology analysis: (**A**) The Venn diagram analysis for DGS and HCD targets. (**B**) PPI network of the DGS compounds targets against CAD. (**B’**) MCODE analysis of PPI network. (**C**) Biological process analysis of PPI networks with a clustering score of 23.667. * Represents a potentially important biological process. (**D**) KEGG enrichment analysis of PPI networks with a clustering score of 23.667. * Represents a potentially important signaling pathway.

**Figure 3 molecules-27-04075-f003:**
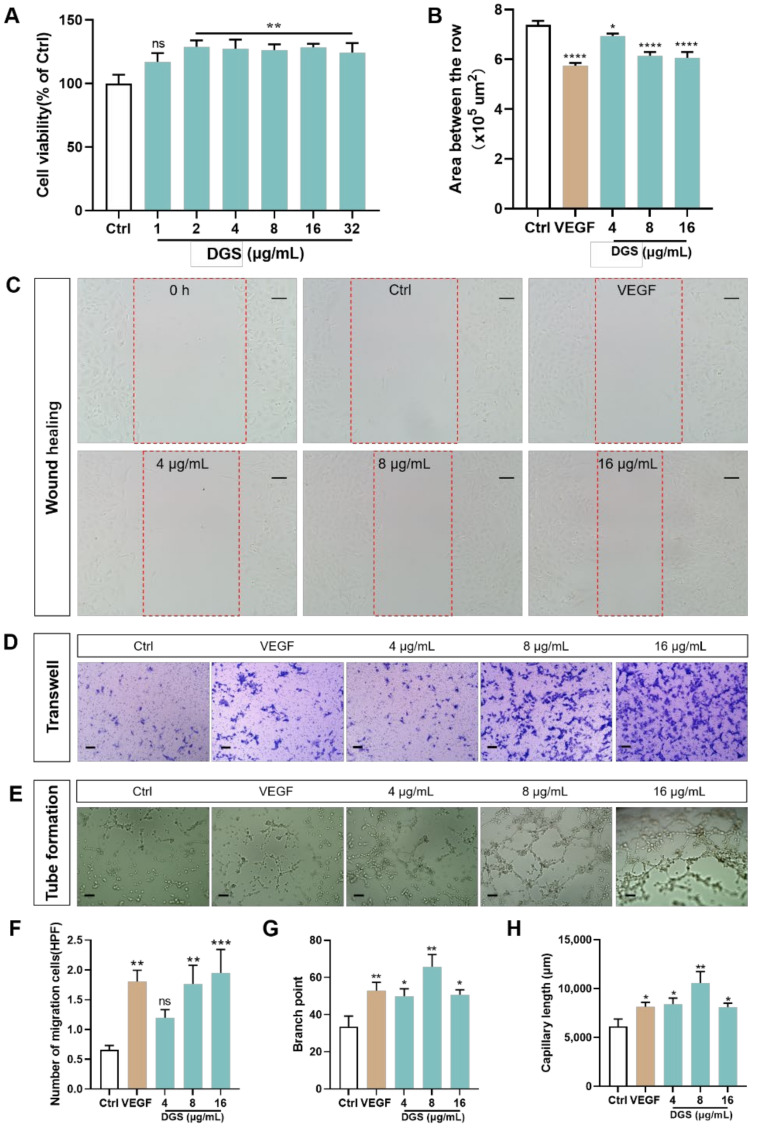
Effect of DGS on HUVECs cells in vitro: (**A**) An CCK-8 assay was carried out to measure HUVECs viability. (**B**) Effect of different concentrations of DGS on the migration of HUVECs cells. Results are presented as the mean ± SEM. (**C**) The healing area of the wound at 0 and 24 h were photographed by microscopy. The red dashed box represents the area counted after migration. Scale bar: 100 μm. (**D**) The migration of HUVECs in Transwell migration assays. Scale bar: 100 μm. (**E**) DGS promoted tube formation of HUVECs. Scale bar: 100 μm. (**F**) Quantification of the number of migrated cells. (**G**) Quantitative analysis of branch points for tube formation assays. (**H**) Quantitative analysis of capillary length for tube formation assays. Values are expressed as the mean ± SEM. ^ns^
*p* < 0.05 vs. Control, * *p* < 0.05 vs. Control, ** *p* < 0.01 vs. Control, *** *p* < 0.001 vs. Control, **** *p* < 0.0001 vs. Control.

**Figure 4 molecules-27-04075-f004:**
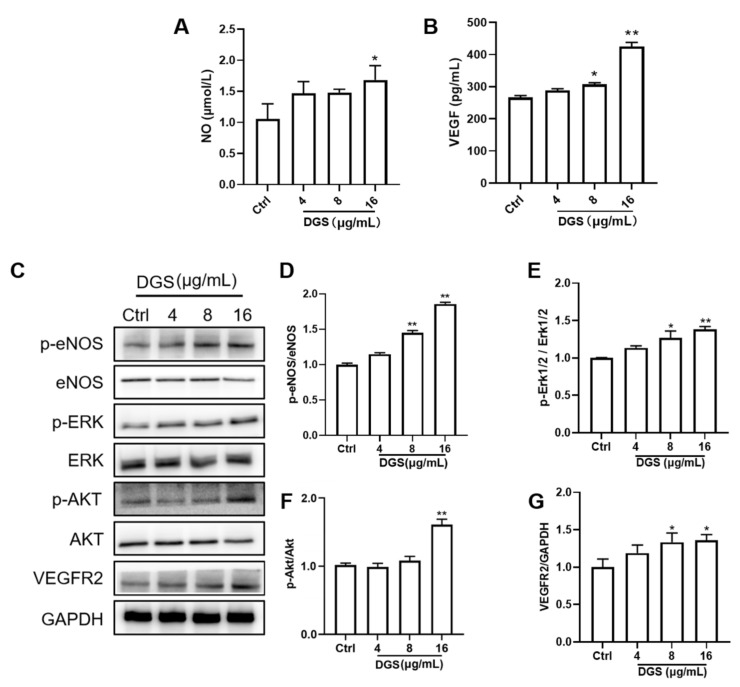
Regulation of NO, VEGF, and related proteins by DGS: (**A**) Effect of DGS on NO levels. (**B**) Effect of DGS on VEGFA levels. (**C**) Western blot results. (**D**–**G**) are the results of statistical analysis of VEGFR2/GAPDH, p-Akt/Akt, p-Erk1/2/Erk1/2, and p-eNOS/eNOS, respectively. Data are presented as the mean ± SEM from at least three independent experiments. * *p* < 0.05 vs. Control, ** *p* < 0.01 vs. Control.

**Figure 5 molecules-27-04075-f005:**
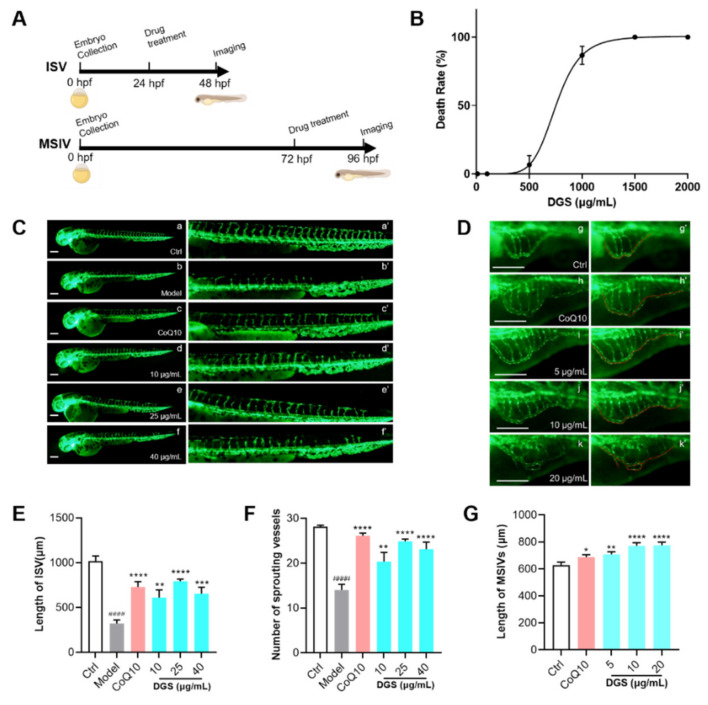
DGS promoted the angiogenesis of zebrafish: (**A**) Schematic diagram of the zebrafish experiment. (**B**) The lethal curve of DGS (**C**) Fluorescent images of the ISV of the zebrafish. The images of a’–f’ were partial enlargements of images a–f. Scale bar: 200 μm. (**D**) Fluorescent image of the MSIV of the zebrafish. The images of g’–k’ were partial enlargements of images g–k. Scale bar: 200 μm. (**E**) Effect of DGS on the length of ISV in zebrafish. (**F**) The effect of DGS on the sprouting of SIV in zebrafish. (**G**) Effect of DGS on the growth of the MSIV in zebrafish. Values are expressed as the mean ± SEM (n = 10). ^####^
*p* < 0.0001 vs. Control, * *p* < 0.05 vs. Model, ** *p* < 0.01 vs. Control, *** *p* < 0.01 vs. Control, **** *p* < 0.0001 vs. Control.

**Figure 6 molecules-27-04075-f006:**
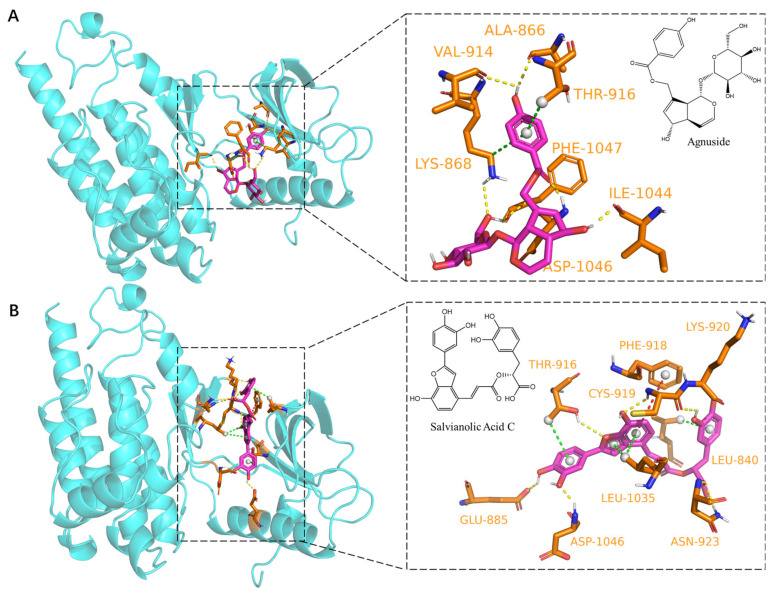
Molecular docking of Agnuside and Salvianolic Acid C to VEGFR2 protein: (**A**) Agnuside binding model with VEGFR2, yellow dashed lines represent hydrogen bonding interactions, green dashed lines represent π–Sigma interactions. (**B**) Salvianolic Acid C binding model with VEGFR2, yellow dashed lines represent hydrogen bonding interactions, green dashed lines represent π–Sigma interactions and red dashed lines represent π–π stacked interactions.

**Figure 7 molecules-27-04075-f007:**
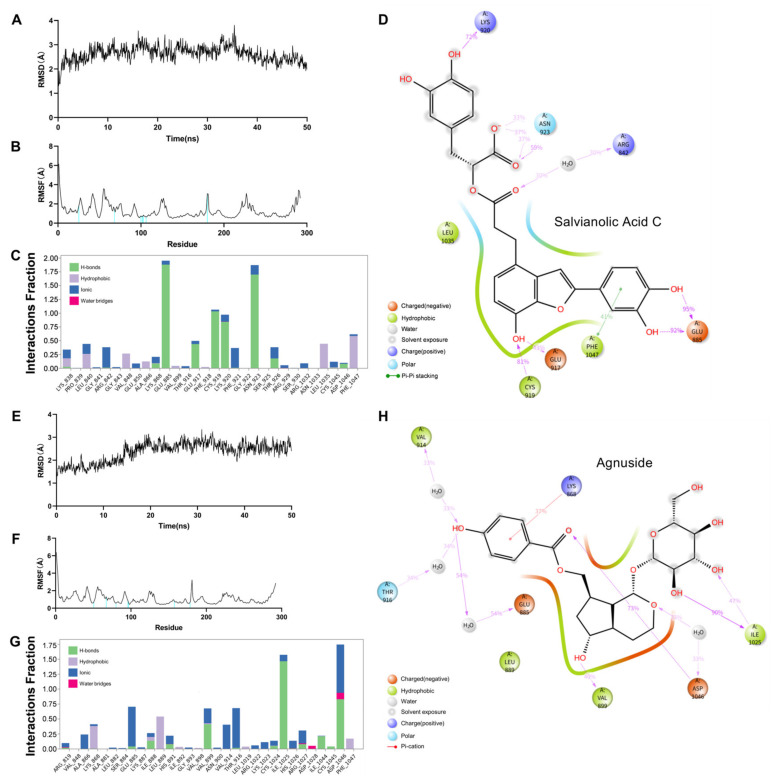
Dynamics molecular docking of two protein–ligand complexes, Salvianolic Acid C–VEGFR2 (**A**–**D**) and Agnuside–VEGFR2 (**E**–**H**): (**A**) RMSD of Salvianolic Acid C–VEGFR2. (**B**) RMSF of Salvianolic Acid C–VEGFR2. (**C**) Protein–Ligand Contacts Diagram of Salvianolic Acid C–VEGFR2; Y axis suggests that percentage of the simulation time the specific interaction is maintained; Values over 1.0 are possible as some protein residue may make multiple contacts of the same subtype with the ligand. (**D**) A schematic of detailed Salvianolic Acid C atom interactions with the VEGFR2 residues. (**E**) RMSD of Agnuside–VEGFR2. (**F**) RMSF of Agnuside–VEGFR2. (**G**) Protein–Ligand Contacts Diagram of Agnuside–VEGFR2. (**H**) A schematic of detailed Agnuside atom interactions with the VEGFR2 residues.

**Figure 8 molecules-27-04075-f008:**
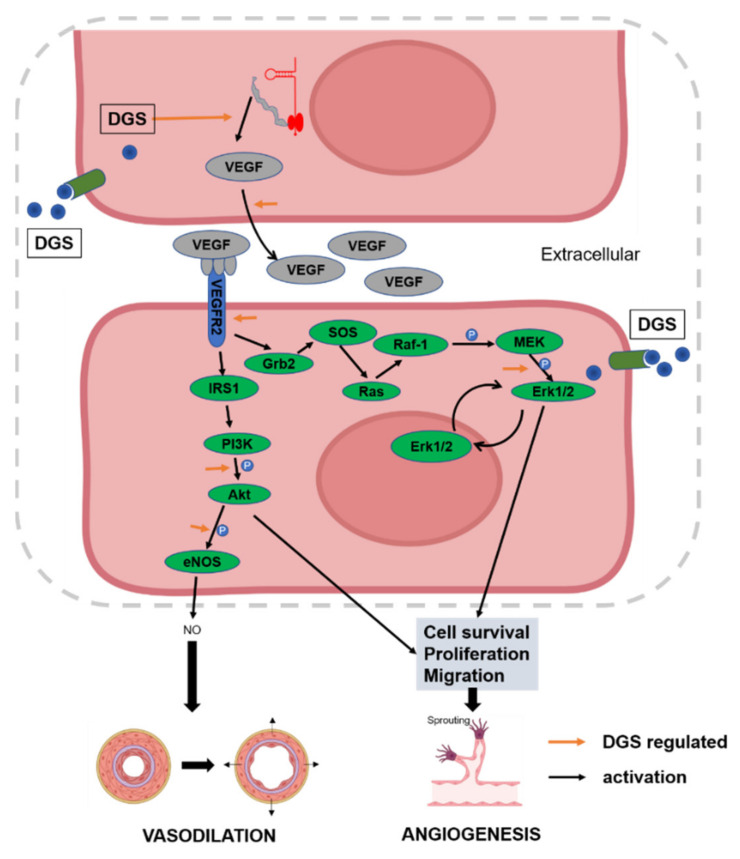
DGS participates in the overall regulatory network of CAD inhibition through angiogenesis and vasodilation.

**Table 1 molecules-27-04075-t001:** Compound analysis and identification of extract of DGS.

Peak No.	Retention Time (min)	Formular	Calc. Mass	Molecular Ion (*m/z*)	Mass Error (ppm)	Fragment Ion	Compound Name
1	1.256	C_7_H_12_O_6_	191.05611	191.0576	7.8	173.0375, 93.0279, 87.0043, 85.0251	Quinic acid
2	3.268	C_11_H_12_O_7_	255.0510	255.0505	−2.1	165.0451, 147.0418, 131.0459	Piscidic acid
3	4.823	C_26_H_28_O_13_	547.14571	547.1468	2.0	295.0490, 267.0542	Mirificin
4	4.908	C_30_H_26_O_12_	577.1352	577.138	3.2	425.5437, 255.4897	Procyanidin B2
5	5.23	C_15_H_14_O_6_	289.07176	289.0737	6.9	205.2053, 189.8794	(−)-Epicatechin
6	6.283	C_21_H_20_O_10_	431.0937	431.0899	−8.8	311.0481, 283.0500	Vitexin
7	6.425	C_16_H_18_O_9_	353.0878	353.08501	−7.9	191.0235, 179.0027	Chlorogenic acid
8	6.866	C_11_H_12_O_6_	239.0561	239.0551	−4.2	179.0277, 177.0486, 107.0453	2-(Carboxymethyl)-4,5-dimethoxybenzoic acid
9	7.199	C_22_H_22_O_11_	461.1035	461.1033	−0.4	253.0397	Tectoridin
10	7.55	C_26_H_28_O_14_	563.1406	563.1428	3.9	311.0433, 283.0471, 227.0628	Schaftoside
11	8.52	C_21_H_20_O_9_	415.1005	415.0963	−7.2	296.0397, 267.0549, 207.0577, 193.0550	Puerarin
12	9.137	C_26_H_20_O_10_	491.0984	491.0985	0.3	363.1897, 351.6277, 320.0439	Salvianolic acid C
13	9.202	C_22_H_22_O_10_	445.1140	445.1103	−8.3	283.0402	3′-Methoxypuerarin
14	10.518	C_15_H_10_O_4_	253.0506	253.0487	−7.6	223.0306, 209.0538	Daidzein
15	11.318	C_29_H_34_O_14_	605.1876	605.1836	−6.6	297.0657, 253.0771	Pueroside A
16	11.706	C_21_H_20_O_10_	431.0984	431.0955	−6.7	269.0337	Genistin
17	12.261	C_27_H_30_O_14_	577.1563	577.1557	−1.1	413.0696, 313.0373, 293.0346	Vitexin 2″-O-rhamnoside
18	13.302	C_15_H_10_O_5_	269.0456	269.0478	8.4	252.0337	Genistein
19	13.341	C_27_H_30_O_15_	593.1512	593.1483	−4.9	311.0412, 283.0351	Tectorigenin 7-O-xylosylglucoside
20	14.927	C_21_H_20_O_12_	463.0882	463.0851	−6.7	303.0245	Hyperoside
21	15.64	C_27_H_30_O_16_	609.1461	609.1415	−7.6	301.03118	Rutin
22	15.74	C_18_H_16_O_8_	359.0772	359.0761	−3.2	161.0169, 135.0388, 119.4798	Rosmarinic acid
23	16.1	C_26_H_22_O_10_	493.1140	493.1100	−8.2	295.0498, 313.0585, 197.0355, 162.0195	Salvianolic acid A
24	16.452	C_27_H_22_O_12_	537.1038	537.0986	−9.8	313.0583, 295.0497	Lithospermic acid
25	17.822	C_36_H_30_O_16_	717.14611	717.1420	−5.7	429.1065, 339.0366, 320.0352, 279.0567, 185.0106	Salvianolic acid B
26	19.44	C_27_H_30_O_15_	593.1512	593.1471	−6.9	414.0734, 311.0541, 293.0322	Vitexin-4″-O-glucoside
27	20.929	C_36_H_30_O_16_	717.1461	717.1420	−5.7	537.0832, 493.0698, 339.0386, 295.0556	Salvianolic acid L
28	23.049	C_29_H_26_O_12_	565.1352	565.1376	4.3	339.0376, 321.0272, 293.0328	Dimethyl Lithospermate
29	24.749	C_26_H_20_O_10_	491.0984	491.0939	−9.1	295.0532	isosalvianolic acid C
30	31.259	C_20_H_28_O_2_	301.2162	301.2156	−2.1	271.6406, 259.5924	Sugiol
31	33.13	C_16_H_16_O_5_	287.0925	287.0906	−6.8	269.2039, 258.1443	Shikonin
32	33.32	C_19_H_18_O_3_	293.1138	293.1192	3	231.3145, 221.1474	Tanshinone IIA
33	33.845	C_19_H_22_O_4_	313.14453	313.1455	3.0	227.0246, 212.0280, 267.0357	Tanshinone V
34	35.93	C_19_H_20_O_3_	295.1340	295.1362	7.6	277.0811, 265.0752, 209.0584	Cryptotanshinone
35	36.129	C_19_H_22_O_3_	297.1496	297.1519	7.7	270.3796, 253.1184	2-[2-(6-methoxy-3,4-dihydro-2H-naphthalen-1-ylidene)ethyl]-2-methylcyclopentane-1,3-dione
36	36.555	C_17_H_14_O_6_	313.0717	313.0703	−4.4	295.1204, 283.1224, 268, 255.1049	Salvianolic acid F
37	37.433	C_39_H_54_O_7_	633.3797	633.3828	4.9	617.3838, 471.358	3-O-p-Coumaroyltormentic acid

**Table 2 molecules-27-04075-t002:** Docking information of VEGFR2 with the corresponding compounds.

Ligand	Binding Affinity (kcal/mol)	Type of Interaction
Agnuside	−8.8	Hydrogen bonding: PHE-1047, ILE-1044, ASP-046, LYS-868, VAL-914, ALA-866;
		π–Sigma: THR-916
Salvianolic Acid C	−10.7	Hydrogen bonding: LYS-920, CYS-919, ASN-923, ASP-1046, GLU-885, THR-916;
		π–Sigma: LEU-840, LEU-1035, THR-916;
		π–π stacked: PHE-918
Isosalvianolic Acid C	−9.7	Hydrogen bonding: GLU-917, CYS-919, ASN-923;
		π–Sigma: PHE-1047;
		π–π stacked: PHE-918, PHE-1047
Genistin	−9.6	Hydrogen bonding: GLU-885, THR-916, ASN-923;
		π–Sigma: THR-916, VAL-848, LEU-1035;
		π–π stacked: PHE-918
3-O-p-Coumaroyltormentic acid	-9.4	Hydrogen bonding: ILE-1025;
		π–π stacked: PHE-845
Tanshinone IIA	−9.2	π–Sigma: PHE-1047, PHE-845;
		π–π stacked: PHE-845
2-[2-(6-Methoxy-3,4-Dihydro-2H-Naphthalen-1-Ylidene)Ethyl]-2-Methylcyclopentane-1,3-Dione	−9.2	π–Sigma: LEU-840,
		π–π stacked: PHE-845
Daidzein	−9.1	π–Sigma: THR-916, LEU-1035, LEU-840;
		π–π stacked: PHE-918
Rosmarinic Acid	−9	Hydrogen bonding: GLU-917, CYS-919, ASP-1046, LYS-868;
		π–Sigma: LEU-889, LEU-1035
Lithospermic Acid	−9	Hydrogen bonding: GLU-885, MET-869, ASP-1046;
		π–π stacked: PHE-845

## Data Availability

The data presented in this study are available on request from the corresponding author.

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
