# Peer review of "Investigation Driven by Network Pharmacology on Potential Components and Mechanism of DGS, a Natural Vasoprotective Combination, for the Phytotherapy of Coronary Artery Disease"

_molecules, 2022, doi:10.3390/molecules27134075_

Round 1

Reviewer 1 Report

The authors evaluated effect of DGS on coronary artery disease by using in vitro, in vivo and in silico models. They found that the DGS extract, consist of Salviae Miltiorrhizae Radix et Rhizoma (Danshen), Puerariae Lobatae Radix (Gegen) and Crataegi Folium (Shanzhaye) exerted angiogenesis and vasodilation effect by activating VEGF and its downstream pathway. The study covers extensive data to present valuable candidate of an herbal prescription to treat CAD.

However, some major points need to be addressed to publish this manuscript.  

  1. Overall, the quantity of the data is okay. But these data are not intimately connected to each other. I think whole bunch of Figure 3 should be transferred into posterior part (change it as figure 5, after in vitro study).
  2. English writing skill is poor throughout the manuscript. I suspect if the authors received grammatical correction service. English correction process is strongly recommended for non-native speakers. Authors should proofread their manuscript very carefully before the submission.
  3. Graphical abstract presented in figure 8 looks crude. The quality of figure should be improved.
  4. Proper references were not cited in the method part. When authors cited web databases or open source softwares, it is necessary to cite proper reference as the developer of the database guides. For instance, ‘Nature Protocols 2009; 4(1):44 & Nucleic Acids Res. 2009;37(1):1’ should be cited for ‘DAVID database’. This should be applied to all other sources.
  1. In figure 5C and D-G, it is better to present quantification result of western blot after normalization of intensity. Adjust number of ‘ctrl group’ as ‘1’ for convenience of readers.
  2. In Figure 4E, the image is not clear. It looks like the focus is not on the identical levels between groups (control, VEGF, 4, 8, 16 ug/ml). The resolution of figure should be improved either.
  3. In Figure 2D, the fold enrichment figure of ‘VEGF signaling pathway’ is missing; It must have been deleted by mistake or the reason should be addressed by authors.  
  4. In figure 1 LC-qTOF/MS data, tick labeling of Y-axis should be modified by referring to other good articles available (Number looks not tidy. How about writing it in horizontal direction with smaller font size?). Also, unit is missing in Y-axis title.
  5. In Table1, What is it mean of PPM? It is better to write ‘Mass error’ or ‘Error‘ in top column combined with ‘PPM’.
  6. In Table 1, Mass error figures seem too high. For reliability of LC/MS data, accuracy below 5ppm is needed. The compounds with issue are as follows; Vitexin (-8.8), Chlorogenic acid (-7.9), Puerarin (-7.2), 3'-Methoxypuerarin (-8.3), Lithospermic acid (-9.8), isosalvianolic acid C (-9.1), Cryptotanshinone (7.6). How do authors explain this error?
  7. Figure 5, I suggest the order of bar chart (5D-G, blot quantification) should be consistent to the sequence of the blot image (5C).
  8. Author did not describe the detailed condition of HUVEC cell culture for western blot (cell incubation time, duration of sample treatment etc).

Other minor points;

Line 32 what is it means of ‘relative systematic network molecular mechanism’? Did author mean ‘molecular mechanism of DGS based on systematic network analysis’? Author should clear the meaning of the sentence.  

Line 57-59. This is not scientific description of biological/pharmacological activity of three herbs. Explanations based on traditional medicine should be avoided for non-alternative medicine journals.

Line 67 Meaning of sentence is awkward. Please correct it.

Line 102 ‘Venn analysis’ -> ‘Venn diagram analysis’?

Line 135 ‘Zebrafish assay is requested’ please correct it.

Line 138 The whole sentence of ‘the efficacy assessment of DGS~’ needs to be revised.

Line 506, 515 What is the exact model of Olympus inverted microscope? And their nationality of company and location?

There are many other typographical, spacing, and grammatical errors in addition to the lists above. Some sentences are not clear with their meaning. I ask authors check prepositions, adverbs, tense of verbs very carefully.

Reviewer 2 Report

This is a well-written manuscript on the potential components and pharmacodynamic mechanisms of DGS for the intervention of Coronary Artery Disease. The manuscript's organisation has been organised easy for readers to follow and catch the information.

I have only a question. Danshen–Gegen (DG) decoction is prescribed to treat coronary heart disease in Chinese medicine. The extract with the optimal ratio of Danshen to Gegen (7:3, w/w), is prepared in boiling water (Phytomedicine, 2011, 18, 916-925).

Why did you use the plant ratio: Danshen: Gegen: Hawthorn leaf = 1:1:1?

Why did you decide to extract the DGS in 80% ethanol and not water, which is the reported solvent? Have you done any tests of the DGC decoction?

Round 2

Reviewer 1 Report

Dear authors,

All my questions have been addressed. I think the manuscript is qualified to be published in the journal. I wish authors all the best.

just one thing.

In line 89, what is NoMB database? Adding an explanation/citation will be appreciated.